# Comparison between femoral block and PENG block in femoral neck fractures: A cohort study

Céline Allard[1], Emmanuel Pardo[1,2], Christophe de la Jonquière[1], Anne Wyniecki[1], Anne Soulier[1], Annibal Faddoul[1], Eileen S. Tsai[3], Francis Bonnet[1,2], Franck Verdonk[1,2,3]*

1 Department of Anesthesiology and Intensive Care, Hôpital Saint-Antoine, Assistance Publique-Hôpitaux de Paris, Paris, France, 2 GRC 29, DMU DREAM, Assistance Publique-Hôpitaux de Paris, Sorbonne University, Paris, France, 3 Department of Anesthesiology, Perioperative and Pain Medicine, Stanford University School of Medicine, Stanford, California, United States of America

* fverdonk@stanford.edu

## Abstract

### Background

Regional analgesia is worth performing in the multimodal postoperative management of hip fracture (HF) because it reduces hospital morbidity and mortality. The aim of this study is to compare the efficacy and side effects of the recently described "Pericapsular Nerve Group (PENG) Block" with those of the femoral block, which is considered the standard of care for postoperative pain control after femoral neck fracture.

### Materials and methods

We conducted a comparative observational study at a university hospital (Saint Antoine Hospital, Sorbonne University, Paris, France), where the PENG block was introduced in August 2019. We include all patients from June to October 2019, who were coming for femoral neck fractures and who had an analgesic femoral block or PENG block before their surgery. The primary outcome was the comparison of cumulative postoperative morphine consumption 48 hours after surgery.

### Results

Demographics, medical charts, and perioperative data of 42 patients were reviewed: 21 patients before (Femoral group) and 21 patients after the introduction of PENG block (PENG group) in clinical practice. Thirteen total hip arthroplasties (THA) and eight hemi arthroplasties (HA) were included in each group. Demographics were also comparable. The median, postoperative, morphine equivalent consumption at 48 hours was 10 [0–20] mg and 20 [0–50] mg in Femoral and PENG groups respectively (p = 0.458). No statistically significant differences were found in postoperative pain intensity, time to ambulation, incidence of morphine-related side effects, or length of hospital stay. The postoperative muscle

**Data Availability Statement:** Data cannot be shared publicly because of PHI contents. Data are available from the Institutional Data Access

"Protection des Donnees de l'APHP" (contact via protectiondesdonnees.aphp6@aphp.fr) for researchers who meet the criteria for access to confidential data.

**Funding:** The authors received no specific funding for this work.

**Competing interests:** The authors have declared that no competing interests exist.

strength of the quadriceps was greater in the PENG group than in the Femoral group (5/5 vs. 2/5, p = 0.001).

## Conclusion

In the management of hip fractures, PENG block is not associated in our study with a significant change in postoperative morphine consumption, compared to femoral block. However, it does significantly improve the immediate mobility of the operated limb, making it appropriate for inclusion in enhanced recovery programs after surgery.

## Introduction

The incidence of femoral neck fracture is 330/10000 per year in Europe. It is increasing every year due to the ageing of the population. By 2050, it is estimated that worldwide, more than 4.5 million patients per year will suffer a femoral neck fracture. [1] The healthcare cost is estimated to be $17 billion per year in the United States, related not only to hospital stay but also to rehabilitation, management of complications, and associated disability [2]. Mortality related to HF in patients aged 55 years and more is estimated between 4% and 16% at one month and between 11% and 43% at one year after surgery [3–5]. Associated morbidity is also important with a significant loss of autonomy at one year in 50% of patients [6].

A recent Cochrane review pointed out a reduced risk of postoperative complications associated with the use of regional anesthesia (RA), a shorter delay to mobilization, and a reduced morphine consumption in hip fractures, reinforcing the value of postoperative pain control as part of a multimodal management [7].

Innervation of the hip joint depends on the femoral, the obturator, and the lateral cutaneous nerves of thigh. None of the RA technique is able to block simultaneously these three nerves and their corresponding endings, even the "3-in-1" block which can theoretically block all the three cited nerves by only one injection does not often reach the territory of the obturator nerve [8]. We usually consider the femoral nerve block as one of a standard of care for hip fractures [7]. But this bloc relieves pain only 69% of the patients with HF after 30 min [9]. In addition, it induces a motor block of the quadriceps muscle after being performed, provoking a potential delay in reeducation. A recent anatomical study shows that the anterior hip joint capsule is mainly innervated by three nerves: the femoral nerve, the obturator nerve, and, inconsistently, the accessory obturator nerve, whose joint branches pass between the iliopubic eminence and the anteroinferior iliac spine [10].

A new block, the "Pericapsular Nerve Group (PENG) block" has been described and initially developed in 2018 by Giron-Arango et al [11]. This diffusion block could provide complete postoperative analgesia with the advantage of sparing the motor function of quadriceps. Case reports have described the efficacy of this block but no study has comparatively evaluated the efficacy of PENG block versus femoral block [12–14].

The main objective of this study was to evaluate the first 48 postoperative hours of morphine consumption before and after the introduction of the PENG block for patients with femoral neck fractures in our institution. The secondary objectives were to compare the motor function of quadriceps muscle, the time to ambulation and the incidence of side effects related to morphine consumption.

## Materials and methods

### Details of study design

In our institution, since August 1$^{st}$ 2019, the PENG block has been used to patients scheduled for hip arthroplasty for femoral neck fracture (total hip arthroplasty (THA) and hemi arthroplasty (HA)) as the cornerstone of postoperative analgesic protocols, in combination with systemic analgesia. The standard of care was previously the use of femoral block associated with systemic non-opioid and opioid analgesics. In order to evaluate this practice, this observational, monocentric, data-based, comparative study compared analgesic effect and functional consequences of these two types of blocks over the two periods defined as "before" and "after" the PENG block implementation on consecutive patients who met the inclusion criteria.

### Settings

Before the PENG block technique was introduced, all the anesthesiologists of the department received a theoretical and practical formation in an expert center during July 2019. We constituted two groups of patients: the "femoral block" (patients operated between June 1$^{st}$ 2019 and August 1$^{st}$ 2019 with a preoperative femoral block) and the "PENG block" study groups (between August 1$^{st}$ 2019 and October 31 2019).

### Sample size estimation

The number of subjects to be included was calculated based on data from a local cohort from our center evaluating the postoperative characteristics of patients who underwent surgery for a fracture of the proximal femur. These data showed an average morphine consumption at 48 hours of 22 milligrams (± 8mg standard deviation). With regard to the literature [15, 16] published on the benefit of a locoregional analgesia in a context of hip fracture, we anticipated a 30% decrease in morphine consumption at 48 hours with the realization of a PENG block compared to a femoral block. With an alpha risk at 0.05 and a power of 80%, inclusion of 21 patients in each group (total number of 42 patients) was considered as sufficient to observe a significant difference between both groups.

### Patients

Eligible patients were aged 18 years or older, were admitted for hip arthroplasty (THA or HA) for femoral neck fracture, and had undergone a preoperative RA technique (femoral block or PENG block). The exclusion criteria were patients with chronic pain before surgery (taking opioids), patients with multiple trauma, patients who could not assess pain reliably (dementia), and patients who had surgery under spinal or epidural anesthesia.

### Data collection

Information related to early postoperative pain (analgesics consumed and dose, visual analog scale (VAS) on arrival and two hours after surgery (H2)) and functional recovery of the operated limb (mobility of the quadriceps defined by the medical research council (MRC) [17] scale rated from 0 to 5) have been collected. During the following postoperative period (from discharge from recovery room (RR) until 48h postoperatively), data collected concerned postoperative pain and postoperative rehabilitation (time to ambulation and length of hospital stay). All opioids administered in the early and late postoperative period were converted to oral morphine equivalent in mg. Data were collected using the electronic health records.

## Primary and secondary outcomes

The primary outcome was the difference in cumulative morphine consumption at 48 hours after surgery. The secondary outcomes were postoperative VAS scores at 2 hours (H2), 12 hours (H12), 24 hours (H24) and 48 hours (H48) after surgery, proportion of VAS $\geq$ 4 at H24 and H48, immediate postoperative mobility of the operated limb quadriceps as defined by the Medical Research Council (MRC) scale [17] before discharge from RR (2 hours after the end of surgery), delay before first step, incidence of side effects related to the administration of morphine within 48 hours postoperatively and length of hospitalization.

## Intervention

Briefly, patients were installed in the pre-anesthesia room and monitored. The blocks were performed by an experienced anesthesiologist, using a Mindray (Shenzen, China) model TE7 ultrasound scanner with a low frequency probe (5 MHz) for the PENG block (**Figs 1 and 2**) and a high frequency probe (12 MHz) for the femoral nerve block.

The PENG block is a diffusion block on the anterior surface of the hip joint capsule. The target nerve branches of this block are the articular branches of the femoral nerve, the obturator nerve and the accessory obturator nerve, leading to no quadricipital motor block. Using a curvilinear low-frequency ultrasound probe (5 MHz) placed in a transverse plane over the Anterior Inferior Iliac Spine (AIIS) and rotated to be aligned with the pubic ramus, a 80 mm RA needle is inserted to inject 20ml of ropivacaine 3.5 mg/ml. The landmarks of this block are externally the AIIS, internally the iliopubic eminence, the psoas tendon anteriorly and the pubic ramus posteriorly."

## Regulatory and ethical aspects

In accordance with French law on biomedical research, this observational study obtained the approval of the Institutional Review Board "Comité d'Ethique de la Recherche en Anesthésie-Réanimation (CERAR, president Prof J.E. Bazin, May 6th 2020) under the reference number IRB 00010254–2020–078.

In order to guarantee the security of personal data, the investigators collected and integrated the information anonymously into a secure database in accordance with the French Commission Nationale de l'Informatique et des Libertés (CNIL) MR-004 methodology, and registered in the Assistance Publique–Hôpitaux de Paris (AP-HP) processing register under the number 20200512144042.

Patients consented via the websites of the institution (AP-HP and Hôpital Saint-Antoine) to the possible use of their data in research aimed towards improving the quality of care, and were informed about their rights and terms of objection. This information was also included for each patient in the hospital's welcome booklet, which was given during administrative registration and presented at the end of the hospitalization reports.

## Statistical methods

The variables were compared between the groups initially by univariate analysis, by Fischer and Chi$^2$ exact tests for the qualitative variables and by Mann-Whitney or Student's t-test according to the normality of the quantitative variables evaluated by a Shapiro-Wilk test. The tests were bilateral, with an alpha risk of $\alpha = 0.05$. Results were given as median (25th-75th percentiles) for quantitative variables and as numbers of patients (percentage proportion) for qualitative variables. A p-value of less than 0.05 was considered significant.

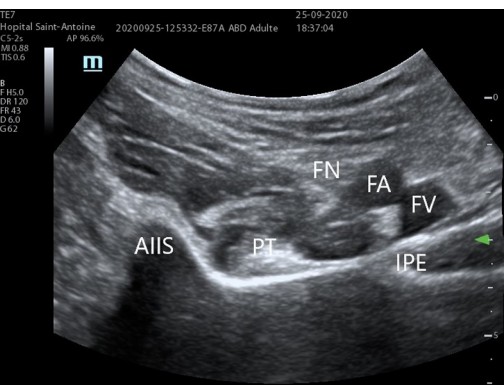

**Fig 1. Small axis ultrasound view of the PENG block.** Using a 5 MHz abdominal probe. Images from St. Antoine Hospital, Paris.

The results are presented in the form of a box-plot diagram. The rectangle extends from the first quartile to the third quartile and is intersected by the median. The segments at the ends of the rectangle provide information on the extreme values.

The R Software (version 3.6.2 for Macintosh, GNU GPL licenses, The R foundation for statistical computing, Vienna, Austria) and the Rstudio interface Version 1.2.5033 (Boston, USA) were used to perform the statistical analyses.

## Results

A total of 42 consecutive patients were included: 21 in the "Femoral block" group and 21 in the "PENG block" group. This population was predominantly female (61.9%, 26 patients), with a median age of 80.5 years Interquartile range (IQR) (70.2, 86.7). Preoperative characteristics (age, gender, ASA, weight/height, co-morbidities) were comparable in both groups. Surgical characteristics in each group were also comparable. There were 8 HA and 13 THA in each group. The surgical approaches performed were comparable between the two groups (**Table 1**).

No difference in cumulative morphine consumption at 48 hours was documented between the two groups (20 (0, 50) mg vs 10 (0, 20) mg, p = 0.478), in the PENG block and the femoral block groups respectively) (**Fig 3**).

The postoperative pain measured by the VAS showed no difference between the two groups, whatever the time of evaluation (**Fig 4**).

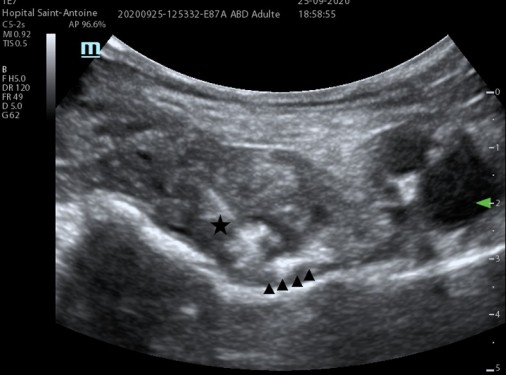

**Fig 2. PENG block realization.** Needle (★) approaching the psoas tendon. ▲ represents injection zone under the psoas tendon. AIIS: Anterior inferior iliac spine, PT: Psoas tendon, IPE: Iliopubic eminence, FA: Femoral Artery, FV: Femoral Vein.

**Table 1. Patients demographics and surgical characteristics of patients in both groups.**

| | Femoral block | PENG block | p-value |
|---|---|---|---|
| Sample size, n | 21 | 21 | |
| Sociodemographic Characteristics | | | |
| Age (IQR) in years | 77.0 (68.0, 84.0) | 83.0 (71.0, 88.0) | 0.38 |
| Male, n (%) | 5 (23.8) | 11 (52.4) | 0.11 |
| BMI (IQR) in kg/m$^2$ | 22.0 (20.0, 24.0) | 23.1 (20.8, 24.9) | 0.53 |
| Number of patients with AGIRG score > 3, n (%) | 18 (85.7) | 15 (71.4) | 0.45 |
| Surgical characteristics | | | |
| ASA classification, n (%) | | | |
| 1 | 3 (14.3) | 3 (14.3) | 1.00 |
| 2 | 9 (42.9) | 10 (47.6) | |
| 3 | 9 (42.9) | 8 (38.1) | |
| Preoperative hemoglobin (IQR) in g/dl | 13.6 (12.0, 14.3) | 12.0 (11.4, 13.4) | 0.14 |
| Time between trauma and surgery in hours (IQR) | 39.0 (25.5, 46.5) | 49.0 (25.0, 96.0) | 0.25 |
| Preoperative VAS score (IQR) | 4.5 (3.0, 8.5) | 5.0 (3.0, 7.0) | 0.61 |
| Number of patients who received preoperative morphine, n (%) | 11 (52.4) | 9 (42.9) | 0.76 |
| Dose of preoperative morphine in mg (IQR) | 18.0 (0.0, 50.0) | 0.0 (0.0, 24.0) | 0.15 |
| Number of patients who received stage 1 and/or 2 analgesics, n (%) | 21 (100.0) | 20 (95.2) | 1.00 |
| Type of surgery | | | |
| HA, n (%) | 8 (38.1) | 8 (38.1) | 1.00 |
| THA, n (%) | 13 (61.9) | 13 (61.9) | |
| Surgical approach | | | |
| Anterior, n (%) | 12 (57.1) | 13 (61.9) | 0.43 |
| Posterior, n (%) | 8 (38.1) | 5 (23.8) | |
| Lateral, n (%) | 1 (4.8) | 3 (14.3) | |
| Administered doses of sufentanil in μg (IQR) | 20.0 (15.0, 25.0) | 20.0 (20.0, 25.0) | 0.99 |
| Operating time in minutes (IQR) | 101.0 (100.0, 120.0) | 95.5 (88.5, 111.3) | 0.17 |

Abbreviations: ASA, American Society of Anesthesiologist; AGIRG, Autonomy Gerontology Iso-Ressources Group; IQR, InterQuartile Range; VAS, Visual Analog Scale.

a = values are given as median 95% CI or percentages.

b = p-value compares Femoral block group versus PENG block group.

c = Fisher test was used to compare qualitative variable, t-test for quantitative variable.

Twenty-four hours after surgery, two (9%) patients in the "PENG block" group reported having a VAS $\geq$ 4, versus three (14%) patients in the "femoral block" group (p = 1.00).

Forty-eight hours after surgery, two (9%) patients in the "PENG block" group had an VAS $\geq$ 4 versus two (9%) patients in the "femoral block" group (p = 1.00).

Regarding the inferior limb motricity, there is a statistically significant difference between the two groups with a median MRC score mobility in the PENG block group rated at 5 IQR (4, 5) which means normal muscle strength of the quadriceps of the operated hip versus a median score of 2 IQR (2, 3.8) in the femoral block group (p<0.001).

There is no statistically significant difference in the time to first step, which is a median of two days (2, 3), between the groups (p = 0.88). The median length of hospitalization was 12 days (7, 12.5) in the PENG Block group versus 11 days (7, 15) in the femoral block group (p = 0.69).

There were 4 cases of Acute urine retention in the PENG block group versus 3 in the femoral block group (p = 1.00) and 2 cases of confusion in the femoral block group versus 6 in the

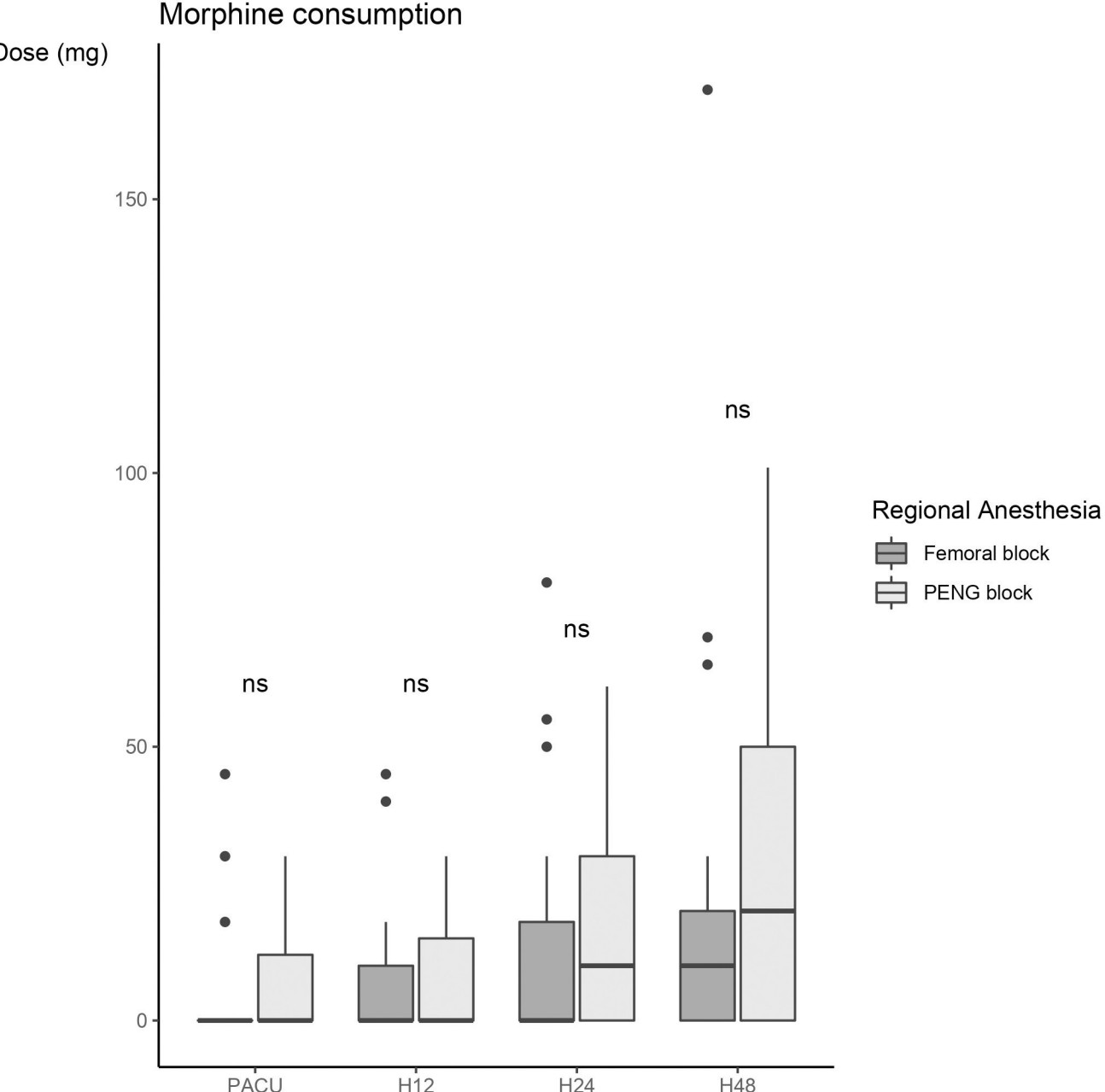

**Fig 3. Morphine consumption at different times: Before leaving the RR, 12 hours postoperatively (H12), 24 hours postoperatively (H24) and 48 hours postoperatively (H48).** Explanation of the figure: Box-plot diagram: The rectangle extends from the first quartile to the third quartile and is intersected by the median. The segments at the ends of the rectangle provide information on the extreme values.

PENG block group (p = 0.24). No statistically significant difference was observed in the incidence of these side effects.

## Discussion

### Main findings of the present study

This study documented that PENG block is comparable to femoral block for postoperative pain control but produces less quadriceps muscle block. The PENG block has been described

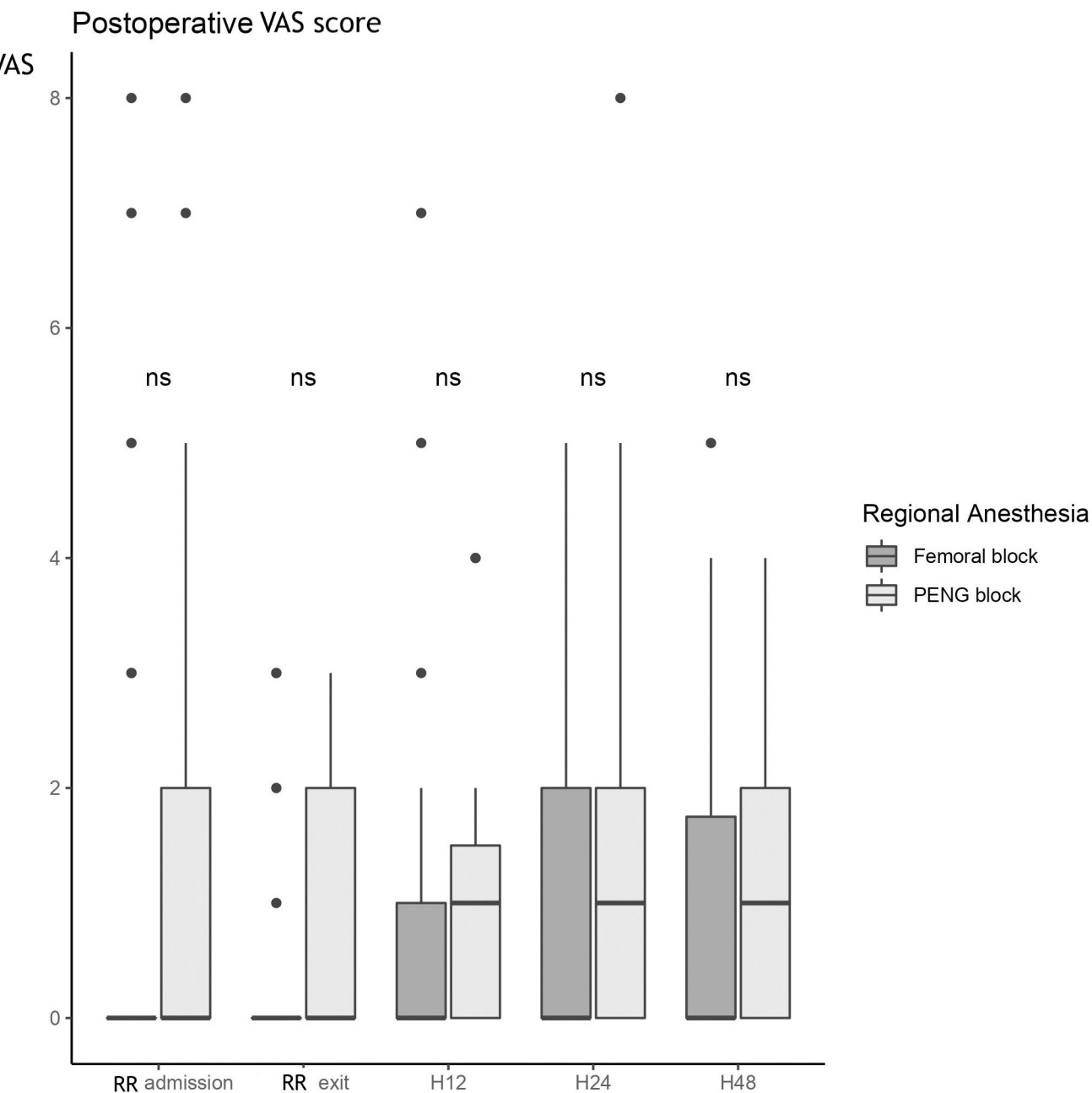

**Fig 4. Postoperative VAS score at different times: Arriving in RR, before leaving the RR, 12 hours postoperatively (H12), 24 hours postoperatively (H24) and 48 hours postoperatively (H48).** Explanation of the figure: Box-plot diagram: The rectangle extends from the first quartile to the third quartile and is intersected by the median. The segments at the ends of the rectangle provide information on the extreme values.

very first in 2018 for preoperative management of femoral neck fracture [11]. We observed no significant difference in cumulative morphine consumption at 48 hours after surgery between the two groups. Similarly, no statistically significant difference was also found in pain scores, length of stay, or incidence of morphine-related adverse events. However, the immediate mobility of the operated limb quadriceps as measured by the MRC scale was different between the two groups, with an average mobility rating of 5 in the PENG block group versus 2 in the femoral block group (p < 0.001).

To our knowledge, this study is the first one to compare PENG block with another block (here, femoral block) in femoral neck fracture.

## Implication and explanation of findings

Interestingly, while the analgesia area covered by the PENG block is theoretically larger than the one covered by the femoral block, by blocking other nerves involved in hip innervation such as the joint branches of the obturator and accessory obturator nerves, postoperative morphine consumption was not statistically different between the two groups. This can be explained by some points. First, the time from arrival at the emergency department on suspicion of a femoral neck fracture, and the surgery is particularly short in our center thanks to a dedicated care line for these patients, which is a critical point in limiting morphine requirements [18]. Da Costa et al. report a median time between emergency department management and surgery of 24.7 hours IQR (18.8,37.8) in our institution [19], which is significantly lower than what is observed in other countries (32.7 hours) [20]. Second, the mean morphine consumption is twice lower in the current study compared to previous studies [21]. This could be related to the population here described, as one of the exclusion criteria was severe dementia, which implies an impossibility to autonomously and reliably evaluate pain. A meta-analysis by Moschinski et al. [22] reports that postoperative morphine consumption could be significantly different in demented patients compared to non-demented. Finally, one of the other potential reasons for the absence of a statistically significant difference on the primary endpoint concerns the way the blocks were performed. Still, few studies relate the execution modalities of the PENG block as the possibility of a benefit from a continuous RA injection. Moreover, the local anesthetic (LA) solution used in the original PENG block study [11] consisted of 20 ml of bupivacaine, 2.5 mg/ml adrenaline, or 20 ml ropivacaine and 5 mg/ml adrenaline combined with dexamethasone. However, in the present study, the LA used was 20 ml ropivacaine at 3.5 mg/ml and without any adjuvants that could have prolonged the duration of postoperative analgesia.

One of the major theorical advantages of the PENG block—compared to femoral block—confirmed by this study is the preservation of quadricipital motricity of the operated limb from the immediate postoperative period. This is due to the diffusion zone of the PENG block, which includes only the articular branches of the femoral nerve, the obturator nerve, and the accessory obturator nerve [11], thus excluding the motor branches of the femoral nerve, which are anesthetized by the femoral block and are responsible for quadricipital motor innervation. In a previous study, Ghodki et al showed that only 13% of the 30 patients receiving femoral nerve block present a normal quadricipital motricity 12 hours after surgery and, more interestingly, that 17% of them still have muscle weakness 24 hours after surgery [23]. At the opposite, the PENG block, by allowing an intact mobility in the immediate postoperative period of the operated limb, would be synonymous with a reduction of the time to first step and even of the length of stay. This has not been demonstrated in this study, probably due to a lack of early mobilization by physiotherapists in our institution, in patients operated on with an HA or THA until 48 hours.

## Limitations of the study

Finally, one of the limitations of this study is its monocentric and databased design and interoperator variability, which therefore potentially leads to inconsistency in the efficacy of the RA performed, thus diminishing the power of the study. In the study by Mistry et al [24], it is reported that the injection zone of the local anesthesia (LA) in the PENG block is very important and would influence the effectiveness of the block. Optimal injection is reflected in the ultrasound vision of a medial diffusion towards the iliopubic eminence.

Even if the sample size was statiscally calculated, another limitation of the study is the small number of patients.

## Conclusion

In conclusion, PENG block may provide equivalent postoperative morphine consumption, compared to femoral block in patients operated on for femoral neck fractures. However, post-operative quadricipital mobility is significantly preserved with PENG block, which could be exploited by an earlier lifting in the context of ERAS. Further studies with a prospective multi-centric design and with a greater number of patients should be done to confirm this result.

## Acknowledgments

We would like to thank all the anesthesiologists, surgeons, nurses and patients who partici-pated in the realization of this work.

## Author Contributions

**Conceptualization:** Céline Allard, Franck Verdonk.

**Data curation:** Céline Allard.

**Formal analysis:** Céline Allard, Emmanuel Pardo.

**Investigation:** Christophe de la Jonquière, Anne Wyniecki, Anne Soulier, Annibal Faddoul.

**Methodology:** Céline Allard, Emmanuel Pardo, Franck Verdonk.

**Project administration:** Francis Bonnet, Franck Verdonk.

**Supervision:** Francis Bonnet, Franck Verdonk.

**Validation:** Francis Bonnet, Franck Verdonk.

**Visualization:** Franck Verdonk.

**Writing – original draft:** Céline Allard.

**Writing – review & editing:** Eileen S. Tsai, Francis Bonnet, Franck Verdonk.

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
