## [Decision Letter · Decision Letter 0]

25 Feb 2021

PONE-D-21-01032

PAF study: Peng block in Emergency Arthroplasty of the Upper Extremity of the Femur : a before/after study

PLOS ONE

Dear Dr. Allard,

Thank you for submitting your manuscript to PLOS ONE. After careful consideration, we feel that it has merit but does not fully meet PLOS ONE’s publication criteria as it currently stands. Therefore, we invite you to submit a revised version of the manuscript that addresses the points raised during the review process.

Please revise accordingly.

We look forward to receiving your revised manuscript.

Kind regards,

Academic Editor

PLOS ONE

Journal Requirements:

2. Please amend either the title on the online submission form (via Edit Submission) or the title in the manuscript so that they are identical.

Reviewers' comments:

Reviewer's Responses to Questions

**Comments to the Author**

1. Is the manuscript technically sound, and do the data support the conclusions?

Reviewer #1: Yes

Reviewer #2: Yes

2. Has the statistical analysis been performed appropriately and rigorously? 

Reviewer #1: Yes

Reviewer #2: Yes

3. Have the authors made all data underlying the findings in their manuscript fully available?

Reviewer #1: Yes

Reviewer #2: Yes

4. Is the manuscript presented in an intelligible fashion and written in standard English?

Reviewer #1: Yes

Reviewer #2: Yes

5. Review Comments to the Author

Reviewer #1: First, I would like to express sincere gratitude to get an opportunity to review the manuscript. The endeavor of the authors is appreciated, as the topic seems to be relevant and promising. The authors have compared PENG block with femoral nerve block for arthroplasty surgery in femoral neck fractures. However, there is some scope for its improvement.

Specific concerns:

1. The title mentions about emergency arthroplasty. Why were such replacement surgery done in emergency basis?

2. It seems the terms ‘before and after’ to be inappropriate. I think it would be better to mention ‘comparison of two procedures’.

3. Kindly mention the principle of the PENG block.

4. Kindly provide Institutional Review Board reference number and date.

5. Was informed written consent taken from the participants of the study?

6. It would be better if study design were clearly mentioned in title, abstract and main text.

Section wise comments

1. Kindly frame title such that it is accurate, informative, descriptive, succinct, simple and specific. The information from title is not convincing.

a. The author may consider modifying the title.

b. Title should contain design of the study.

c. Kindly avoid abbreviations.

d. The words ‘upper extremity’ in needs to be removed. Instead ‘neck of femur fracture would make more sense’.

2. Methods sections is supposed to be core of any study. Here, methods section contains inadequate information. For example, following components for methods section need to be well described.

i. Details of Study design

ii. Setting

iii. Sample size estimation

iv. Sampling technique

v. Participant

vi. Primary and secondary outcome variables with working definition

vii. Intervention/issue of interest (exposure)

viii. Comparison

ix. Ethics and end point

x. Statistical analysis

3. The discussion section needs to be described scientifically. Kindly frame it along the following lines:

a. Main findings of the present study

b. Comparison with other studies

c. Implication and explanation of findings

d. Strengths and limitations

e. Conclusion, recommendation and future directions

Reviewer #2: The research compared the efficacy and side effects of the “Pericapsular Nerve Group (PENG) Block” with those of the femoral block. The study is interesting and the result is encouraging and practical. The limit is the small scale of the sample.

6. PLOS authors have the option to publish the peer review history of their article (what does this mean?). If published, this will include your full peer review and any attached files.

Reviewer #1: **Yes: **Dr Satish Prasad Barnawal

Reviewer #2: No

---

## [Author Response · Author response to Decision Letter 0]

17 Mar 2021

Response to reviewers #1 and #2

Reviewer #1

First, I would like to express sincere gratitude to get an opportunity to review the manuscript. The endeavor of the authors is appreciated, as the topic seems to be relevant and promising. The authors have compared PENG block with femoral nerve block for arthroplasty surgery in femoral neck fractures. However, there is some scope for its improvement.

First of all, we would like to thank the Dr. Satish Prasad Barnawal for spending time to review our manuscript. It is a great honor to read his comments so that our work would be better after revising it. We hope that our changes would suit his expectations.

Sincerely,

Allard Céline, MD and Verdonk Franck, MD, PhD

Specific concerns:

1. The title mentions about emergency arthroplasty. Why were such replacement surgery done in emergency basis?

Our study concerns only arthrosplasties in the femoral neck fracture setting. So that, all surgeries were done in an emergency basis as delayed surgeries (>24 hours) have been associated with an increase in mortality and incidence of pneumonia, myocardial infarction, and pulmonary embolism at 30 days (1,2).

2. It seems the terms ‘before and after’ to be inappropriate. I think it would be better to mention ‘comparison of two procedures’.

Accordingly to this comment, we corrected our manuscript title which is now “Comparison between femoral block and PENG block in femoral neck fractures : a cohort study”.

3. Kindly mention the principle of the PENG block.

We added the principle of the PENG block in the material and methods section, “intervention” paragraph.

“The PENG block is a diffusion block on the anterior surface of the hip joint capsule. The target nerve branches of this block are the articular branches of the femoral nerve, the obturator nerve and the accessory obturator nerve, leading to no quadricipital motor block. Using a curvilinear low-frequency ultrasound probe (5 MHz) placed in a transverse plane over the Anterior Inferior Iliac Spine (AIIS) and rotated to be aligned with the pubic ramus, a 80 mm RA needle is inserted to inject 20ml of ropivacaine 3.5 mg/ml. The landmarks of this block are externally the AIIS, internally the iliopubic eminence, the psoas tendon anteriorly and the pubic ramus posteriorly.”

4. Kindly provide Institutional Review Board reference number and date.

We added a specific section to clarify this ethical point of utmost importance as “Regulatory and Ethical Aspects” in “Material and methods”, as following:

“In accordance with French law on biomedical research, this observational data-based study obtained the approval of the Institutional Review Board “Comité d’Ethique de la Recherche en Anesthésie-Réanimation (CERAR, president Prof J.E. Bazin, May 6th 2020) under the reference number IRB 00010254 - 2020 – 078.

In order to guarantee the security of personal data, the investigators collected and integrated the information anonymously into a secure database in accordance with the French Commission Nationale de l’Informatique et des Libertés (CNIL) MR-004 methodology, and registered in the Assistance Publique – Hôpitaux de Paris (AP-HP) processing register under the number 20200512144042.”

5. Was informed written consent taken from the participants of the study?

Our study is observational and data-based. In accordance with the French law on biomedical research, all participants of the study as well as all patients from the Assistance Publique -Hôpitaux de Paris signed a consent for possible use of their hospitalization data, as it was done here. In addition, information was delivered through the hospital’s welcome booklet, and in all medical reports at the hospital discharge. To clarify this point to the readers of Plos One, we added this information in the “Regulatory and Ethical Aspects” section as following :

“Patients consented via the websites of the institution (AP-HP and Hôpital Saint-Antoine) to the possible use of their data in research aimed towards improving the quality of care, and were informed about their rights and terms of objection. This information was also included for each patient in the hospital’s welcome booklet, which was given during administrative registration and presented at the end of the hospitalization reports.”

6. It would be better if study design were clearly mentioned in title, abstract and main text.

We fully agree with Dr. Satish Prasad Barnawal’s comment. We changed the manuscript so that the study design is much well explained. This study is a cohort study : comparative and single-center.

7. Methods sections is supposed to be core of any study. Here, methods section contains inadequate information.

Accordingly to the Dr. Satish Prasad Barnawal’s comment, we rewrite the methods to improve the manuscript quality. We added some subheadings so that the comprehension of section is clearer.

8. The discussion section needs to be described scientifically. 

We changed the discussion section to be more accurate by adding subheadings.

 

Reviewer #2

The research compared the efficacy and side effects of the “Pericapsular Nerve Group (PENG) Block” with those of the femoral block. The study is interesting and the result is encouraging and practical. The limit is the small scale of the sample.

We would like to thank reviewer #2 for all the time spent to review our work. We hope that this revised version would suit his/her expectations. We fully agree with reviewer #2 that one of the limits of this study is the small sample of population. We found these results encouraging and will certainly conduct another prospective study with a more important number of patients to confirm the latter. 

Sincerely,

Allard Céline, MD and Verdonk Franck, MD, PhD

---

## [Decision Letter · Decision Letter 1]

11 Apr 2021

PONE-D-21-01032R1

Comparison between femoral block and PENG block in femoral neck fractures: a cohort study

PLOS ONE

Dear Dr. Franck,

Thank you for submitting your manuscript to PLOS ONE. After careful consideration, we feel that it has merit but does not fully meet PLOS ONE’s publication criteria as it currently stands. Therefore, we invite you to submit a revised version of the manuscript that addresses the points raised during the review process.

Please revise accordingly. 

We look forward to receiving your revised manuscript.

Kind regards,

Academic Editor

PLOS ONE

Journal Requirements:

Reviewers' comments:

Reviewer's Responses to Questions

**Comments to the Author**

1. If the authors have adequately addressed your comments raised in a previous round of review and you feel that this manuscript is now acceptable for publication, you may indicate that here to bypass the “Comments to the Author” section, enter your conflict of interest statement in the “Confidential to Editor” section, and submit your "Accept" recommendation.

Reviewer #3: All comments have been addressed

Reviewer #4: (No Response)

2. Is the manuscript technically sound, and do the data support the conclusions?

Reviewer #3: Yes

Reviewer #4: Yes

3. Has the statistical analysis been performed appropriately and rigorously? 

Reviewer #3: Yes

Reviewer #4: Yes

4. Have the authors made all data underlying the findings in their manuscript fully available?

Reviewer #3: Yes

Reviewer #4: Yes

5. Is the manuscript presented in an intelligible fashion and written in standard English?

Reviewer #3: Yes

Reviewer #4: No

6. Review Comments to the Author

Reviewer #3: The article is of scientific interest and in line with the aims of the journal. The manuscript restates the authors guidelines and does not require a revision of the English language by a native speaker. The changes made after the reviews have certainly increased the quality of the article. I recommend its publication after minor revision.

KEYWORDS

In order to improve the visibility of the article, do not use keywords already present in the title.

ABSTRACT

Materials and methods: "42 patients were reviewed: 21 patients before (Femoral group) and 21 patients ..." delete all information relating to the included patients and move it to the results section. In the materials and methods section there should be only the inclusion and exclusion criteria and not information on the patients included.

INTRODUCTION

The introduction should be extended.

MATERIALS AND METHODS

"of 21 patients in each group (total number of 42 patients) was" move to the results section.

Statistical analysis was adequately performed-

RESULTS AND DISCUSSION

The results are written very fluently and are adequately argued in the discussion.

LIMITS

Add the small number of patients.

CONCLUSIONS

Invite future studies that resolve the limitations expressed above.

FIGURES

The figures are of good quality.

TABLES

The tables are well made and adequately cover the text.

REFERENCES

The references are recent and the text is adequate.

Reviewer #4: Comment on the manuscript Number PONE-D-21-01032R1

Comparison between femoral block and PENG block in femoral neck fractures: a

cohort study

The idea of the research is interesting for the readers of the journal. The authors compared the effect of 2 types of nerve block ( Femoral nerve block and Pericapsular Nerve Group Block) on the postoperative pain relief and recovery of patient who underwent hip replacement after fracture neck of femur. They used for this multiple parameters as mentioned in the study. They found no significant difference between both nerve blocks except in one parameter which is the quadriceps muscle strength. This parameter was for the advantage of the Pericapsular Nerve Group Block.

The article submitted for me to review was the second version of the article. I acknowledge the improvements that the Authors did on the first version of the manuscript which has met most of the recommendations of the previous reviewers. In spite of this there is still a room for improvement. I have the following recommendation:

• I recommend the authors lit the article being reviewed by an english language specialist.

• Lines 38 and 88 and other locations: The Term “intermediate hip replacements” is strange and not known in orthopedic surgery. Did the authors mean hemi arthroplasty or bipolar arthroplasty? Please explain and I recommend to use the world wide known terminology.

• Line 104: there is nothing called the upper extremity of the femur. Please Replace with fracture of the proximal femur.

• The abbreviation IQR is repeatedly used in the article without mentioning what it does mean. This was mention under one of the tables. Please write it in full text at the first time to appear in the text.

• Some spelling mistakes:

- Line 71: PEricapsular should be "Pericapsular.

- Line 90: Associated to …. Please replace using : Associated with .

- Line 137: an experimented anesthesiologist please replace with experienced anesthesiologist

I recommend to accept the article for publication after considering the previous recommendations.

7. PLOS authors have the option to publish the peer review history of their article (what does this mean?). If published, this will include your full peer review and any attached files.

Reviewer #3: No

Reviewer #4: **Yes: **Dr. Ayman F. AbdelKawi, MD

---

## [Author Response · Author response to Decision Letter 1]

19 Apr 2021

Response to reviewers #3 and #4

Reviewer #3

The article is of scientific interest and in line with the aims of the journal. The manuscript restates the authors guidelines and does not require a revision of the English language by a native speaker. The changes made after the reviews have certainly increased the quality of the article. I recommend its publication after minor revision.

First of all, we would like to thank the reviewer #3 for spending time to review the new version of our manuscript. We hope that our changes would suit his expectations.

Sincerely,

Allard Céline, MD and Verdonk Franck, MD, PhD

Specific concerns:

1. KEYWORDS

In order to improve the visibility of the article, do not use keywords already present in the title.

We deleted the keyword “PENG block” which already appear in the title.

2. ABSTRACT

Materials and methods: "42 patients were reviewed: 21 patients before (Femoral group) and 21 patients ..." delete all information relating to the included patients and move it to the results section. In the materials and methods section there should be only the inclusion and exclusion criteria and not information on the patients included.

We re-wrote this section so that this information appears in the section “results” and not in the “materials and methods” section.

3. INTRODUCTION

The introduction should be extended.

We added some information in the “introduction” paragraph. 

Line 54 “By 2050, it is estimated that more than 4.5 million patients worldwide per year will suffer a femoral neck fracture. (1)”

Line 67 “even the “3-in-1” block which can theoretically block all the three nerves by only one injection does not often reach the territory of the obturator nerve (8).”

4. LIMITS

Add the small number of patients.

This limitation of the study is added in line 296 “Even if the sample size was statistically evaluated, another limitation of the study is the small number of patients.”

5. CONCLUSIONS

Invite future studies that resolve the limitations expressed above.

We added this sentence in the conclusion section in line 304 “Further studies with a prospective multicentric design and with a greater number of patients should be done to confirm these results.” 

Reviewer #4

The idea of the research is interesting for the readers of the journal. The authors compared the effect of 2 types of nerve block (Femoral nerve block and Pericapsular Nerve Group Block) on the postoperative pain relief and recovery of patient who underwent hip replacement after fracture neck of femur. They used for this multiple parameters as mentioned in the study. They found no significant difference between both nerve blocks except in one parameter which is the quadriceps muscle strength. This parameter was for the advantage of the Pericapsular Nerve Group Block.

The article submitted for me to review was the second version of the article. I acknowledge the improvements that the Authors did on the first version of the manuscript which has met most of the recommendations of the previous reviewers. In spite of this there is still a room for improvement. I have the following recommendation:

• I recommend the authors lit the article being reviewed by an english language specialist.

• Lines 38 and 88 and other locations: The Term “intermediate hip replacements” is strange and not known in orthopedic surgery. Did the authors mean hemi arthroplasty or bipolar arthroplasty? Please explain and I recommend to use the world wide known terminology.

• Line 104: there is nothing called the upper extremity of the femur. Please Replace with fracture of the proximal femur.

• The abbreviation IQR is repeatedly used in the article without mentioning what it does mean. This was mention under one of the tables. Please write it in full text at the first time to appear in the text.

• Some spelling mistakes:

- Line 71: PEricapsular should be "Pericapsular.

- Line 90: Associated to …. Please replace using : Associated with .

- Line 137: an experimented anesthesiologist please replace with experienced anesthesiologist

We would like to thank Dr. Ayman F. AbdelKawi, MD for all the time spent to review our work. 

We changed our manuscript by respecting all his remarks and suggestions. We sent the manuscript to an English native speaker (Dr. Eileen Tsai) to check the language. We hope that it will now suit his expectations.

Sincerely,

Allard Céline, MD and Verdonk Franck, MD, PhD

---

## [Decision Letter · Decision Letter 2]

13 May 2021

PONE-D-21-01032R2

Comparison between femoral block and PENG block in femoral neck fractures: a cohort study

PLOS ONE

Dear Dr. Franck,

Thank you for submitting your manuscript to PLOS ONE. After careful consideration, we feel that it has merit but does not fully meet PLOS ONE’s publication criteria as it currently stands. Therefore, we invite you to submit a revised version of the manuscript that addresses the points raised during the review process.

Please revise accordingly.

We look forward to receiving your revised manuscript.

Kind regards,

Academic Editor

PLOS ONE

Journal Requirements:

Reviewers' comments:

Reviewer's Responses to Questions

**Comments to the Author**

1. If the authors have adequately addressed your comments raised in a previous round of review and you feel that this manuscript is now acceptable for publication, you may indicate that here to bypass the “Comments to the Author” section, enter your conflict of interest statement in the “Confidential to Editor” section, and submit your "Accept" recommendation.

Reviewer #3: All comments have been addressed

Reviewer #5: (No Response)

2. Is the manuscript technically sound, and do the data support the conclusions?

Reviewer #3: Yes

Reviewer #5: Yes

3. Has the statistical analysis been performed appropriately and rigorously? 

Reviewer #3: Yes

Reviewer #5: Yes

4. Have the authors made all data underlying the findings in their manuscript fully available?

Reviewer #3: Yes

Reviewer #5: Yes

5. Is the manuscript presented in an intelligible fashion and written in standard English?

Reviewer #3: Yes

Reviewer #5: Yes

6. Review Comments to the Author

Reviewer #3: I thank the authors for the corrections made. The work has been improved and is now acceptable for publication.

Reviewer #5: This study was a before and after protocol, comparing the phase of femoral block to newly applied PENG block. The methods were simple. Simply, the authors showed non-inferiority of PENG regarding perioperative pain control, and the results seemed to be sound. I would address some minor comments.

Minor comments

#1. Regarding the inclusion criteria, I do not understand how the author managed those who the block was not sufficient, and the technique of analgesia was converted to other ways (general anesthesia or spinal?).

#2. I understand that calculating sample size was so difficult. However, for me, a 30% decrease of morphine consumption in 48 hours seemed to be too large. Reference # 15 and # 16 would not generate the idea of 30% decrease. I do not have the impression that the effects of block last for 48 hours. The pain scores in the surgery or morphine consumption in OR or PACU would have some reason.

#3. The authors showed that the MRC scale before discharge from RR was different. Please wrote the timing of MRC measurement in the result (line 221 to 224). How long the muscle weakness last would be a good discussion point. The description of ERAS in discussion seemed to be over-speculation. It can be deleted.

#4. In the conclusions, the authors wrote that PENG provided “equivalent”. However, because of the small sample size, probably the description would be written in more mild way.

#5. For me, the description of acknowledgment was too broad. Probably, a little more specific description would be favorable.

#6. Regarding figure legends, the publishers will request the title of figure and the explanation.

7. PLOS authors have the option to publish the peer review history of their article (what does this mean?). If published, this will include your full peer review and any attached files.

Reviewer #3: No

Reviewer #5: No

---

## [Author Response · Author response to Decision Letter 2]

16 May 2021

Response to reviewers #3 and #5

Reviewer #3

I thank the authors for the corrections made. The work has been improved and is now acceptable for publication.

Dear Reviewer #3,

Thank you again for reviewing our work. We hope to work with you again soon.

Sincerely yours,

Dr. Céline ALLARD and Dr. Franck VERDONK 

Reviewer #5

This study was a before and after protocol, comparing the phase of femoral block to newly applied PENG block. The methods were simple. Simply, the authors showed non-inferiority of PENG regarding perioperative pain control, and the results seemed to be sound. I would address some minor comments.

Minor comments

#1. Regarding the inclusion criteria, I do not understand how the author managed those who the block was not sufficient, and the technique of analgesia was converted to other ways (general anesthesia or spinal?).

This study is a before and after protocol comparing two types of analgesic block -femoral block and PENG block- before a total hip arthroplasty (THA) or hemi arthroplasty (HA) both performed in emergency for hip fracture. Both analgesic blocks were performed in the pre-anesthesia room, right before entering the operating room. 

All surgeries were done under general anesthesia as written in the methods section. We made the decision, prior to data collection and analysis, to exclude patients operated under spinal anesthesia considering that spinal anesthesia will modify pain and muscle strength evaluation after the surgery, in addition to the potential systemic effect of local anesthetics and morphine in this setting.

#2. I understand that calculating sample size was so difficult. However, for me, a 30% decrease of morphine consumption in 48 hours seemed to be too large. Reference # 15 and # 16 would not generate the idea of 30% decrease. I do not have the impression that the effects of block last for 48 hours. The pain scores in the surgery or morphine consumption in OR or PACU would have some reason.

We totally agree with this comment. However, calculating sample size was very difficult because there is actually no study in the literature comparing morphine consumption after femoral block and another type of analgesic block. References #15 and #16 evoke the possibility of a 30% decrease in morphine consumption. 

We also agree with the reviewer that the effect of analgesic block may not last for more than 24 hours. The primary end point of this study was the cumulative morphine consumption at 48 hours after the surgery as one of the common endpoints in studies focusing on peri-operative medicine 1,2. That being said, the pain scores after surgery recorded in PACU didn’t show any significant difference between the two groups of patients.

#3. The authors showed that the MRC scale before discharge from RR was different. Please wrote the timing of MRC measurement in the result (line 221 to 224). How long the muscle weakness last would be a good discussion point. The description of ERAS in discussion seemed to be over-speculation. It can be deleted.

We thank the reviewer for his comment. We added this sentence in the methods section to explain when the MRC score was measured :

“immediate postoperative mobility of the operated limb quadriceps as defined by the Medical Research Council (MRC) scale (17) before discharge from RR (2 hours after the end of surgery).”

We deleted the paragraph concerning ERAS in the discussion and added this sentence concerning the duration of the muscle weakness after a femoral block :

“In a previous study, Ghodki et al showed that only 13% of the 30 patients receiving femoral nerve block present a normal quadricipital motricity 12 hours after surgery and, more interestingly, that 17% of them still have muscle weakness 24 hours after surgery.”

#4. In the conclusions, the authors wrote that PENG provided “equivalent”. However, because of the small sample size, probably the description would be written in more mild way.

We modified the conclusion to address this comment : 

“In conclusion, in this study, PENG block may provide equivalent postoperative morphine consumption, compared to femoral block in patients operated on for femoral neck fractures.”

#5. For me, the description of acknowledgment was too broad. Probably, a little more specific description would be favorable.

We modified a bit the acknowledgment section :

“We would like to thank all the anesthesiologists, surgeons, nurses and patients who participated in the realization of this work.”

#6. Regarding figure legends, the publishers will request the title of figure and the explanation.

The titles of the figures already are in the manuscript. The figures were uploaded separately in the Plos one online submission form, as it was asked.

We added the explanation of the figure in the manuscript :

“Explanation of the figure : box-plot diagram : the rectangle extends from the first quartile to the third quartile and is intersected by the median. The segments at the ends of the rectangle provide information on the extreme values.”

---

## [Decision Letter · Decision Letter 3]

21 May 2021

Comparison between femoral block and PENG block in femoral neck fractures: a cohort study

PONE-D-21-01032R3

Dear Dr. Franck,

We’re pleased to inform you that your manuscript has been judged scientifically suitable for publication and will be formally accepted for publication once it meets all outstanding technical requirements.

Kind regards,

Academic Editor

PLOS ONE

Additional Editor Comments (optional):

Reviewers' comments:

Reviewer's Responses to Questions

**Comments to the Author**

1. If the authors have adequately addressed your comments raised in a previous round of review and you feel that this manuscript is now acceptable for publication, you may indicate that here to bypass the “Comments to the Author” section, enter your conflict of interest statement in the “Confidential to Editor” section, and submit your "Accept" recommendation.

Reviewer #3: All comments have been addressed

Reviewer #5: All comments have been addressed

2. Is the manuscript technically sound, and do the data support the conclusions?

Reviewer #3: Yes

Reviewer #5: (No Response)

3. Has the statistical analysis been performed appropriately and rigorously? 

Reviewer #3: Yes

Reviewer #5: (No Response)

4. Have the authors made all data underlying the findings in their manuscript fully available?

Reviewer #3: Yes

Reviewer #5: (No Response)

5. Is the manuscript presented in an intelligible fashion and written in standard English?

Reviewer #3: Yes

Reviewer #5: (No Response)

6. Review Comments to the Author

Reviewer #3: (No Response)

Reviewer #5: (No Response)

7. PLOS authors have the option to publish the peer review history of their article (what does this mean?). If published, this will include your full peer review and any attached files.

Reviewer #3: No

Reviewer #5: No

---

## [Editor Report · Acceptance letter]

25 May 2021

PONE-D-21-01032R3 

Comparison between femoral block and PENG block in femoral neck fractures: a cohort study 

Dear Dr. Verdonk:

I'm pleased to inform you that your manuscript has been deemed suitable for publication in PLOS ONE. Congratulations! Your manuscript is now with our production department. 

Kind regards, 

on behalf of

Dr. Robert Jeenchen Chen 

Academic Editor

PLOS ONE